# Traditional Foods, Globalization, Migration, and Public and Planetary Health: The Case of *Tejate*, a Maize and Cacao Beverage in Oaxacalifornia

Daniela Soleri [1],* , David Arthur Cleveland [1,2] , Flavio Aragón Cuevas [3], Violeta Jimenez [4] and May C. Wang [4]

1   Department of Geography, University of California, Santa Barbara, CA 93106-4060, USA
2   Environmental Studies Program, University of California, Santa Barbara, CA 93106-4160, USA
3   Instituto Nacional de Investigaciones Forestales, Agrícolas y Pecuarias, Villa de Etla, Oaxaca 68200, Mexico
4   Department of Community Health Sciences, UCLA Fielding School of Public Health, Los Angeles, CA 90095, USA
*   Correspondence: soleri@ucsb.edu

**Abstract:** We are in the midst of an unprecedented public and planetary health crisis. A major driver of this crisis is the current nutrition transition—a product of globalization and powerful multinational food corporations promoting industrial agriculture and the consumption of environmentally destructive and unhealthy ultra-processed and other foods. This has led to unhealthy food environments and a pandemic of diet-related noncommunicable diseases, as well as negative impacts on the biophysical environment, biodiversity, climate, and economic equity. Among migrants from the global south to the global north, this nutrition transition is often visible as dietary acculturation. Yet some communities are defying the transition through selective resistance to globalization by recreating their traditional foods in their new home, and seeking crop species and varieties customarily used in their preparation. These communities include Zapotec migrants from the Central Valleys of the southern Mexican state of Oaxaca living in greater Los Angeles, California. Focusing on the traditional and culturally emblematic beverage *tejate*, we review data from our research and the literature to outline key questions about the role of traditional foods in addressing the public and planetary health crisis. We conclude that to answer these questions, a transnational collaborative research partnership between community members and scientists is needed. This could reorient public and planetary health work to be more equitable, participatory, and effective by supporting a positive role for traditional foods and minimizing their harms.

**Keywords:** maize genetic diversity; nutrition transition; Oaxaca; Mexico; Oaxacalifornia; planetary health; public health; sugar sweetened beverages; *tejate*; traditional beverages; traditional food; Zapotec culture

## 1. Introduction

We are in the midst of an unprecedented public and planetary health crisis—a pandemic of obesity and diet-related non-communicable diseases (NCDs), as well as industrial agriculture's negative impact on the biophysical environment, biodiversity, climate, and economic equity, that threatens the supply and accessibility of food itself [1,2]. A major driver of this crisis is the current nutrition transition—a product of globalization and powerful multinational food corporations promoting the increased production and consumption of environmentally destructive, relatively unhealthy, and ultra-processed foods, which replace healthier and more environmentally sustainable foods [3–6].

This transition results in the ongoing loss of traditional small-scale farmers and the crop genetic diversity they maintain in situ in their fields. Traditional agroecological farming systems and the farmer-maintained crop genetic diversity within them are vital sources of diversity for these farmers in adapting to ongoing climate, environmental, and

socioeconomic changes that characterize the public and planetary health crisis, as well as for the development of new crop varieties and agroecological practices globally [7].

Because Mexico is a global center of diversity of many crops, including maize, conserving diversity in situ is critically important [8]. A key cause of the loss of crop diversity in Mexico has been the Green Revolution, beginning in the mid-20th century, including the decision by "development" professionals to focus on larger-scale, industrial agriculture, and modern crop varieties dependent on chemical fertilizers, and not on small-scale maize farmers and their traditional varieties [9]. The Mexican government continued in this direction with its neoliberal economic and agricultural policies, especially since the 1980s, with the North American Free Trade Agreement (NAFTA, or the Tratado de Libre Comercio de América del Norte, TLCAN), which went into effect in 1993 [10–12].

Between 1991 and 2007, NAFTA led to the loss of 4.9 million small-scale ("family") farm jobs in Mexico because small-scale farmers were not be able to compete with subsidized US production, a result predicted by NAFTA proponents [13,14]. NAFTA further integrated the Mexican food system into the global industrial food system, often with traditional, healthy foods replaced by unhealthy, ultra-processed foods [11]. The United States–Mexico–Canada Agreement (USMCA) replaced NAFTA in 2020, and continues NAFTA's provision for tariff and quota free importation of US maize into Mexico [15], making it likely that the effects on small-scale Mexican maize farmers driven by NAFTA will continue.

In addition to these policies, increasing population density, environmental degradation, and climate change have made it more and more difficult for small-scale Mexican maize farmers to continue farming, including in Oaxaca, as they themselves have noted. In a survey of 200 farming families in four Oaxacan Central Valleys communities, 99.5% grew maize, yet there was a decline of 5.3–33.3% (mean = 17.5%) in total kilograms of maize seed farmers reported sowing between 1987 and 2007 [16]. Over 90% of interviewees hoped that their children would continue farming maize and there was a significant positive association between maize varietal richness and farmers' belief that their children would continue to farm. Still, only 48% of farmers believed the next generation actually would continue to farm maize [16].

## 2. Sugar Sweetened Beverages, Public and Planetary Health and the Case of *Tejate*

As mentioned, NAFTA has opened the Mexican economy to US food corporations and a flood of unhealthy, ultra-processed foods, including sugar sweetened beverages (SSBs), a major source of increased added sugar in the diet [17]. In 2015, Mexico had the second highest per capita SSB consumption globally (167 L per year) after the US [18]. There is a well-established association between SSB consumption and the risk of NCDs such as diabetes [19], including for Mexicans [20]. The prevalence of diet-related NCDs in Mexico, including diabetes, has been rising dramatically in recent decades [21,22], and NCDs increase the risk of severe COVID-19 symptoms and death [22].

In response to the public health costs of SSBs, there have recently been increased efforts to implement policies to reduce SSB intake, with taxes being the most popular. Mexico implemented a tax on SSBs in 2014 despite strong industry opposition [23], following a campaign by public health advocates to influence government and the public [24]. The tax has been successful in reducing SSB consumption [25]. In the US, the city of Berkeley, California, implemented the first SSB tax in 2015 [26], which has had some success in reducing SSB consumption [27], with most tax revenues used to support community health programs [28]. San Francisco, Oakland, and Albany, California, also implemented SSB taxes before the state-wide preemption on such taxes became law in 2018.

The current Mexican government has also taken steps to address food system issues more broadly. For example in early 2019 it created the Intersectoral Group on Health, Food, Environment and Competitiveness (Grupo Intersectorial de Salud, Alimentación, Medio Ambiente y Competitividad, GISAMAC) with representatives of diverse government bodies, including secretaries of labor, agriculture, economy, public health, the environment, education, indigenous peoples, biosecurity, and science and technology. The formation of

GISAMAC has raised hopes that comprehensive approaches to the food system will be possible, including making health and nutrition key objectives [29].

In immigrant communities, the nutrition transition is often visible as the abandonment of relatively healthy traditional foods in favor of the less healthy foods that dominate in their destination. This has been documented for Mexican migrants to the US from rural Mexico, with school-age children more likely to adopt new, less healthy, standard American (US) diets [30], which are more likely to result in NCDs compared to traditional Mexican diets [31]. This trend may be exacerbated by the perception among some migrants that traditional Mexican foods are unhealthy compared to standard "American" foods [32].

Yet this transition is not always homogeneous, or complete, and migrants may try to recreate traditional foods, including beverages, in their new homes. *Tejate*, an ancient maize and cacao beverage from the Central Valleys of Oaxaca (CV) is one of these traditional beverages [33,34]. Even while it is being replaced by commercial SSBs such as soda in some rural Oaxacan households, *tejate* is also being re-created by Oaxacan migrants in the greater Los Angeles (LA) area of southern California [35], an indication of its significance to them. By recreating traditional foods, migrants are resisting common features of globalization, such as cultural assimilation and the abandonment of traditional cuisines, crop varieties, and farming [35], all of which are features of globalization that can contribute to the public and planetary health crisis.

However, because today *tejate* is most often served with added sugar, it could be defined as an SSB, and its consumption therefore discouraged by public health programs and policies that include taxes on SSBs or added sugar. In addition, other regulations affecting SSBs could also affect *tejate*, e.g., the City of Berkeley adopted a policy in 2021 prohibiting the city from buying SSBs or serving SSBs on city property, and it has been recommended that SSBs not be sold or served at events of organizations that receive city funding [36].

Because foods and their consumption influence both public and planetary health, we present existing evidence important for public health programs to consider regarding the public and planetary health effects of traditional Mexican beverages that contain maize and other ingredients, and increasingly include added sugar, using the example of *tejate*. These beverages present a challenge for research and practice because of the contrast between their potential health harm from added sugar, and their potential to promote public and planetary health through their contributions to nutrition, culture, social relationships, maize genetic diversity, and traditional agroecology. We discuss existing data on the nutritional, cultural, social, and agricultural roles of *tejate*, and propose questions useful for incorporating traditional beverages and foods into more culturally sensitive and holistic public and planetary health messaging.

### 3. *Tejate*'s Cultural, Social, and Agroecological Roles in Oaxaca

*Tejate* is emblematic of the Zapotec culture in the CV [37], and part of a family of traditional foam-topped beverages found across Mesoamerica made from maize and cacao, with other herbs and plant additives sometimes included in their recipes [33,38–40].

Foam-topped cacao beverages have long been culturally and socially significant in Mesoamerica, as shown by archaeological and historical evidence of their association with ceremonies for major life transitions. Some archeologists have speculated that the exotic ingredients and labor intensive preparation and serving of frothed cacao-based beverages in Mesoamerica as early as the Mid-Formative period (1000–400 BCE) imbued those beverages with great social value [38]. The performative aspect of preparing and serving these beverages—e.g., pouring one liquid into another from a height to produce foam—may also have made them especially valuable for display rituals incurring social debt [38]. Paleobotanical research has found maize and cacao residues in vessels used for beverages from the Postclassic period (800–1521 CE) in the CV, although there is currently no evidence for the presence or absence of other *tejate* ingredients because diagnostic markers have not yet been developed [41]. *Tejate*'s cultural, healing, and celebratory uses persist in the CV today [37].

The significance of cacao-based beverages such as *tejate* is also evidenced by their inclusion in a newly constructed traditional Mexican diet index that considers nutritional, cultural, culinary, and environmental factors to identify key foods or groups of foods that are essential components of this traditional diet [42].

*Tejate* preparation is labor- and skill-intensive. Today, women (exclusively in our experience and in published sources) in Zapotec CV communities prepare *tejate* using maize nixtamalized with wood ashes (*cuanextle*), and a toasted and ground mixture of *cacao rojo* (*Theobroma cacao* L.), *rosita de cacao* blossoms (*Quararibea funebris* (La Llave) Vischer (Bombacea)), mamey seed or *pixtle* (*Pouteria sapota* (Jacq.) H.E. Moore & Stearn), and, in some areas, *cacao blanco* (*T. bicolor* Humb. & Bonpl.), *cocoyul* (*Acrocomia aculeata* (Jacq.) Lodd. ex Mart.), or other more widely known ingredients such as coconut, peanut, or walnut [33]. *Tejate* is served in a *jícara*, a decorated, bowl-shaped vessel made from *Crescentia cujete* L. or *C. alata* (Figure 1).

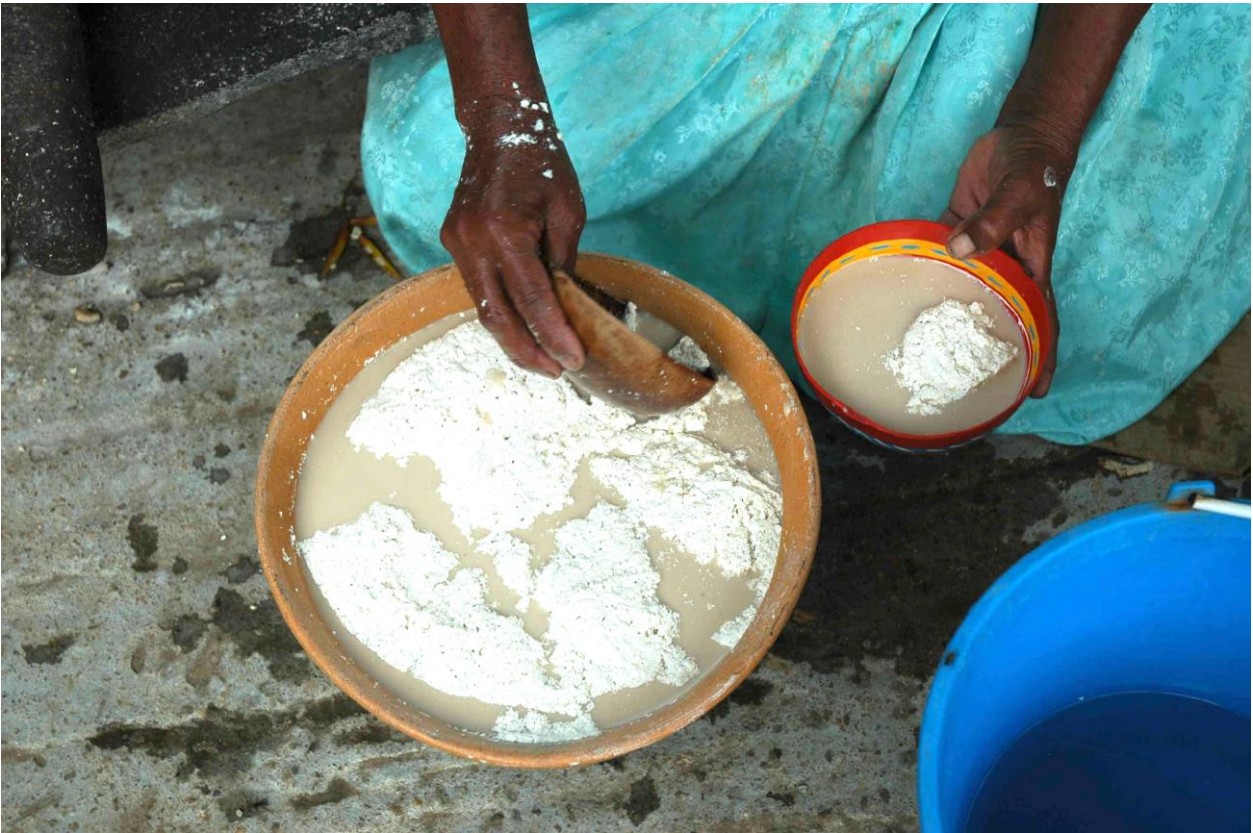

**Figure 1.** Serving *tejate* into a *jícara*. Photo by D Soleri, used with permission of photographer.

Mexico is the center of maize domestication and the primary center of that crop's diversity [43,44], with Oaxaca being home to 60% of the country's 59 native maize races [8]. Across Mexico, these native races, and the hundreds of locally developed traditional maize varieties selected by farmers from them, provide multiple benefits: agroecological (e.g., adaptation to local environmental stresses [45]), crop diversity conservation [8], cultural [46], culinary, and economic [47]. As with other genetically diverse crop populations [48], the diversity present within traditional maize populations has the potential to make them better able to respond to ongoing environmental changes, including anthropogenic climate change [49], requiring fewer external inputs that are themselves often harmful to the climate and environment. A small but growing number of case studies (e.g., [50,51]) provide evidence that traditionally-based agriculture and the biological and social systems it supports can guide the transition to agricultural production that supports instead of undermines planetary health [52]. Similarly, a growing number of agricultural

researchers argue that biodiverse agroecology is the production system best suited to respond to the public and planetary health crisis and other challenges to the Anthropocene food system, including inequity [52,53].

Traditional maize varieties in the CV are predominantly from the race Bolita and include multiple kernel colors and season lengths [8]. Traditional varieties of maize are preferred for agroecological, culinary, and sociocultural reasons in Oaxaca where these varieties are sown on an estimated 85–90% of the area under maize cultivation [8] (FAC field notes, 2022).

In the CV, small-scale maize farmers, cooks, and consumers of traditional maize foods, including *tejate,* strongly prefer that these be made with Bolita varieties, because of Bolita's superior gustatory and health qualities [35,54]. As documented elsewhere in Mexico, farmers may value their traditional varieties so much that they maintain those varieties, even while reducing their total area sown to maize due to deteriorating growing conditions [55]. Despite CV and Mexican communities more generally valuing the quality of tortillas and other foods made from traditional varieties, Mexican government policies favor large-scale production of industrial maize varieties, and target small-scale farmers with cash payment programs that encourage consumption of purchased, industrial maize [56]. Even so, in some locations, purchased tortillas may be more costly despite being lower quality than ones made by households from their own-grown maize [56]. Thus, farm households navigate the demands of both food security and food quality, based on their local circumstances, as we found in our research in the CV [16,33,40].

In 2007 we conducted interviews about *tejate* in three CV communities, with 25 households in each, randomly selected from the community health clinic register. Our interviews documented households' self-reported production and consumption of maize and *tejate* and related household practices such as preparation and procurement [35,54]. Across communities, we found that households prioritized the use of traditional maize varieties for *tejate* over other important traditional maize-based staples, some explicitly stating they would use industrial maize for other foods, but not for *tejate*—with only 2% of households having actually made *tejate* with industrial maize. Earlier research found a positive association between *tejate* preparation and traditional maize diversity maintained by the household [33].

*Tejate* is associated with agricultural work, especially the strenuous physical labor of maize field preparation and harvest [33], and in addition to cash, *tejate* is an expected part of the payments to day laborers (*mozos*) hired for field work. The beverage is also considered an important part of some celebrations and Christian religious festivities including *quinceañeras*, baptisms, and Easter [37]. We found variations in the role *tejate* plays across the three communities. In one community only 12% of households described *tejate* as a regular, normal part of their diet, and not solely associated with field work or festivities, while in San Bartolomé Quialana (SBQ), the community with the highest rate of labor migration, and the greatest consumption of maize, this proportion was 96% [54].

Annual mean maize consumption per person for direct human use in Oaxaca, 196 kg (FAC field notes, 2022), is twice as high as the overall Mexican mean, 97.5 kg (calculated from [57]). Our survey found that reliance on maize in self-reported diets in SBQ was even greater, with a mean annual consumption of 213 kg per person, equal to 109% of the estimated Oaxacan per person mean and 218% of the national mean (FAC field notes, 2022) (Table 1).

For the three communities in our survey, *tejate* accounted for a mean of 11% of household direct human maize consumption (not including maize for *tejate* for *mozos*, maize grain fed to household animals, or consumed in the form of purchased tortillas); in SBQ the mean was 17%, with most households preparing it two or more times a week [54].

**Table 1.** Maize and *tejate*: self-reported production and consumption in San Bartolomé Quialana (SBQ), interview responses 2007, n = 25 households. Based on [35].

| Household Maize and *Tejate* [a,b] | Mean | St Dev | Min | Max |
|---|---|---|---|---|
| Number maize varieties grown, household$^{-1}$ | 1.4 | 0.6 | 0 | 3 |
| Direct human maize consumption, kg person$^{-1}$ annually | 213.4 | 129.3 | 0 | 487 |
| Total direct human maize consumption, household$^{-1}$ annually | 1102.3 | 512.1 | 0 | 2190 |
| Kg maize for *tejate* for family, person$^{-1}$ annually | 28.9 | 21.0 | 0.9 | 104 |
| Total kg maize for *tejate* for family, household$^{-1}$ annually | 148.0 | 88.1 | 6.5 | 365 |
| Total kg maize for *tejate* for mozos, household$^{-1}$ annually | 18.6 | 8.9 | 4.0 | 36 |
| Grand total kg maize for *tejate* annually | 181.1 | 102.5 | 22.5 | 381 |
| Percent of total direct human maize consumption for *tejate*, household$^{-1}$ annually | 17.4% | 14.3% | 0.4% | 57.0% |
| Frequency consuming *tejate*, annually [c] | 211.7 | 109.9 | 104 | 469 |
| Proportion of annual *tejate* consumption prepared at home | 58% | 23% | 0% | 94% |
| Estimate of frequency parents consumed *tejate*, annually [c,d] | 166.4 | 31.8 | 104.0 | 182 |

[a] Does not include maize use for animal feed, or for *mozos* unless otherwise indicated. Does not include kg purchased maize tortillas. [b] Person$^{-1}$ consumption calculated with children as 73% adult male consumption, based on energy consumption, after [58]. [c] Question asked in frequency categories: >1×/week = 1, <1×/week and >1×/month = 2, <1×/month and >1×/year = 3, ~1×/year = 4, never = 5. These were then approximated as the following number of times annually: 1 = 104, 2 = 24, 3 = 6.5, 4 = 1, 5 = 0. [d] We assume this was home prepared as *tejateras* are relatively new in SBQ, see [35].

Even while being valued for both its quotidian and celebratory roles, *tejate*'s place in household life appears to be declining, with surveyed household heads in 2007 stating they prepare *tejate* less frequently than their parents did (see Table 1). A number of respondents pointed out that the laborious and time-consuming preparation is a major reason that some younger people are not continuing to make the beverage. The increased availability of *tejate* sold by *tejate*ras—women who make the beverage in their homes for commercial sale—has countered the reduction in home preparation for some households.

Despite regular *tejate* consumption and *tejate*'s cultural and social importance, all SBQ households in our survey noted that today they are also substituting other beverages for *tejate* in day-to-day use, with sugar-sweetened soda being the most frequently reported substitute (among 72% of households) [54].

### 4. *Tejate* and Nutrition

We estimated *tejate*'s potential nutritional contribution based on *tejate* with 81% water content in a standard *jícara* serving of 355 g, using data from our biochemical and nutritional analysis [34] (Figure 1, Table 2). *Tejate* is clearly a significant element of the diet in SBQ, not only as a prominent feature of the customary cuisine, but also as a source of energy, protein, and fat. In that community, based on a mean total consumption frequency of 211 times annually (Table 1), *tejate* would provide 10% of the requirements for both energy and protein, but likely more, as an instance of *tejate* consumption typically comprises more than one *jícara* for adults in rural households.

*Tejate* was traditionally served *simple* (plain), without added sweeteners, but preparation today typically includes sugar syrup, so it is important to assess the negative health impact of *tejate* consumption. At the same time, we argue, it is also necessary to assess *tejate*'s net effect by considering the potential positive health, cultural, social, and agricultural effects described above, and explore how to enhance benefits and minimize harms.

**Table 2.** Selected dietary contributions of *tejate* and soda, per 355 g (12.5 fl oz) of liquid (equal to one *jícara* serving size of *tejate*) relative to the daily Dietary Reference Intake.

| | | Energy (kcal) | Carbohydrates (g) | Protein (g) | Fat (g) |
|---|---|---|---|---|---|
| Dietary Reference Intake (DRI) | | | | | |
| Mean for adults (female & male), 19–59 years, in southern Mexico [a] | | 1942 | 242 | 41 | 65 |
| *Tejate* | | | | | |
| Mean of four samples collected from households in SBQ [b] | amount | 324.7 | 54.3 | 7.3 | 3.1 |
| | per cent DRI | 16.7% | 22.4% | 17.7% | 4.8% |
| Mean of values for Bolita blanco and amarillo [c] | amount | 288.5 | 51.6 | 5.4 | 7.9 |
| | per cent DRI | 14.9% | 21.3% | 13.1% | 12.1% |
| Soda | | | | | |
| Mexican "Coca-Cola Original" [d] | amount | 108.1 | 26.6 | 0 | 0 |
| | per cent DRI | 5.6% | 11.0% | 0% | 0% |
| US "Coca-Cola Original" [e] | amount | 144.0 | 39.0 | 0 | 0 |
| | per cent DRI | 7.4% | 16.1% | 0% | 0% |

[a] Data from [59], that analyzed intake data from the 2006 Mexican National Health and Nutrition Survey, a nationally representative cross-sectional household survey, including a food frequency questionnaire. Here we use the median intakes for the entire sample (females plus males) for the 10 states of southern Mexico including Oaxaca (n = 6188), which are almost identical to those for rural Mexico. The authors used DRIs from [60], which they adjusted using individual age and sex data for each regional sample of the Mexican population. [b] Data from [34]. *Tejate* samples included a diversity of recipes all containing sugar; 1 made with white Bolita, 1 with yellow Bolita, 2 with yellow and white Bolita combined. [c] Data from [61]. Samples were laboratory prepared *tejate* without added sugar. [d] Composition based on data for 100 mL of Coca-Cola Original in Mexico, from Coca-Cola México (https://www.coca-colamexico.com.mx/nuestras-marcas/coca-cola/coca-cola; accessed on 25 March 2021), calculated for 355 g. The only other nutrient in this beverage is sodium (25.8 mg per 355 g). [e] Composition based on data for 20 oz of Coca-Cola Original in the US (https://us.coca-cola.com/products/coca-cola/original; accessed on 25 March 2021), calculated for 355 g. The only other nutrient in this beverage is sodium (45 mg per 355 g).

Studies of the biochemical and nutritional characteristics of *tejate* support the comments of regular consumers that the beverage not only quenches thirst, but also provides satiety and nutrients in addition to energy, unlike soda and other commercial SSBs that are its most common replacement (Table 2) [33]. Sotelo et al. [34] found that, depending on the recipe, *tejate* can be a source not only of energy, protein, and fat, but also of methylxanthines and minerals, including potassium, iron, zinc, magnesium, and calcium, also identified by another study [61]. The ash nixtamalization of maize used for *tejate,* compared to lime-based nixtamalization of maize for tortillas, tamales, and other foods, is the source of the higher levels of the minerals iron, zinc, and magnesium in *tejate* [62].

In addition to sugar, many of the calories in *tejate* are from starches in maize, a proportion of which are resistant starch (RS) (see below and Table 2, including values for sugarless *tejate*). Lime-based nixtamalization of maize is known to increase RS in preparation of tortillas [63] and tamales [64], and there is evidence for the association of RS with improved metabolic indicators for the risk of obesity, diabetes, and cardiovascular disease (CVD) [65,66], including via effects on the gut microbiome [67].

RS has not been well studied in *tejate*, but one study found that the combination of nixtamalized white Bolita maize and cacao produced a starch-lipid complex (resistant starch 5, or RS5) contributing to its low glycemic index (GI) of 32.7 without added sugar, and 38.2 with added sugar [61], which is much lower than US Coca-Cola (63) [68], in spite of *tejate* having more than twice as many calories (Table 2).

Beverages such as diet soda, which replace added sugar with non-nutritive sweeteners (NNS) are promoted as eliminating the adverse health effects of added sugar. However, consumption of artificial NNSs has been associated with increased risk of disease, including type-2-diabetes [19], obesity, and CVD [69]. The effects of artificial NNSs (e.g., sucralose) compared with natural NNSs (e.g., stevia, *Stevia rebaudiana*) has not been well researched. Replacing added sugar with natural NNSs might reduce *tejate*'s GI (e.g., agave syrup has

a lower GI than sucrose), and stevia has no caloric content [70]. Stevia is increasingly cultivated in Mexico with commercial forms being promoted there [71]. The gastronomic and health effects of replacing sugar with these alternative sweeteners in *tejate* need to be investigated. In the meantime, reducing added sugars would be an initial step in reducing health risks of *tejate* consumption.

### 5. Oaxacalifornia: Migration of People and Foods to California from the Central Valleys of Oaxaca

The CV have experienced high rates of labor migration to the US, including to California, which together with Oaxaca itself is sometimes referred to as Oaxacalifornia, a transnational geographic space and the support networks and values that sustain it [72,73]. In greater LA the large diaspora of CV Zapotecs is re-creating elements of their culture, including *tejate* and other traditional foods (Figure 2), in their new home, in spite of the skill and labor required, and difficulties in obtaining the traditional ingredients, including maize varieties.

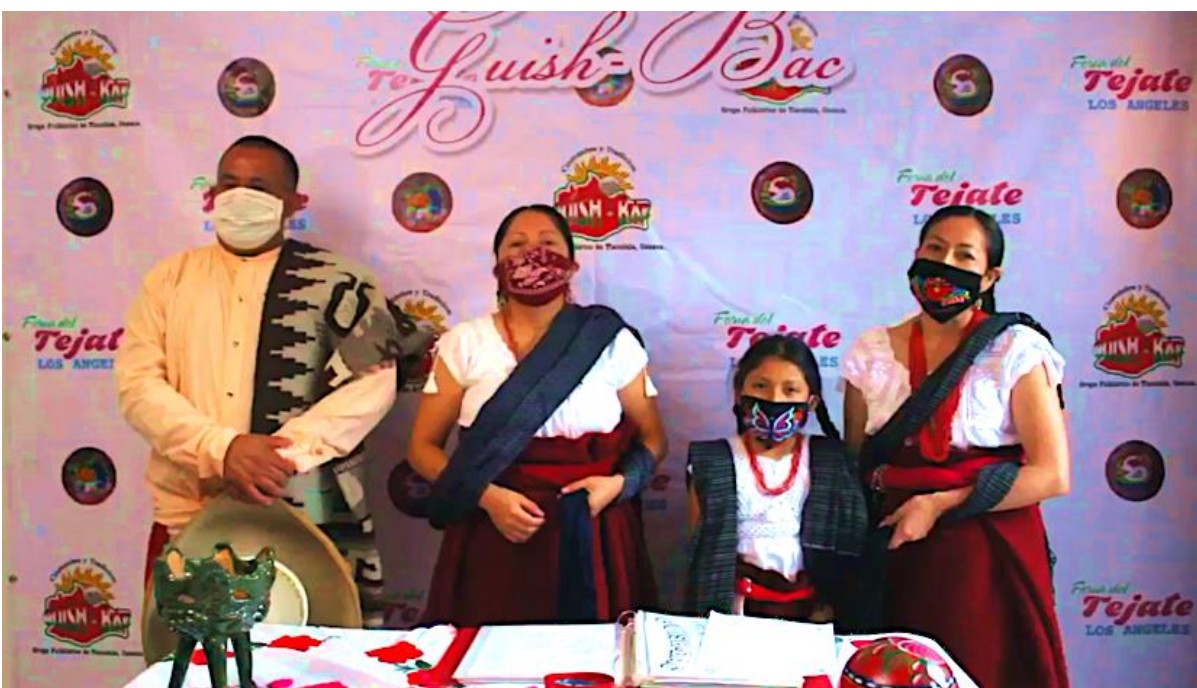

**Figure 2.** Representatives of the Grupo Folklórico Guish-Bac announce the cancellation of the annual *Feria del Tejate* in September 2020 due to the coronavirus pandemic and thank past participants. Credit: Grupo Folklórico Guish-Bac Facebook, used with permission.

### 5.1. Migration, Diet Change, and Health

While the current number of Oaxacan Zapotecs living in greater LA is difficult to determine, estimates suggest there were over 200,000 in the early 2000s [74]. When rural Mexicans migrate to the US there is a documented trend of transitioning from a relatively healthier traditional diet with more fruits, vegetables, whole grains, and legumes, to the less healthy standard US diet, including increased consumption of SSBs and other sources of added sugar [75], and there is a greater likelihood that a standard US diet will result in NCDs compared to traditional Mexican diets [31]. School-age children are more likely to adopt standard US diets [30], hypothesized to be due to social pressures and seeking conformity in school food environments. Oaxacan immigrants' households are often low income, with all adults working, making relatively time and labor intensive traditional foods difficult to maintain, and the ultra-processed food environment difficult to avoid. Some mothers lament their children's increased preference for these foods over more traditional rural Oaxacan diets, when those mothers recognize the traditional diets as

healthier [76]. Yet there is strong interest among Mexican migrants and their descendants in maintaining their traditional foods. This is true even among those who perceive these foods to be less healthy than "American" food, although this perception may be due in part to less healthy versions of traditional foods being preferentially retained by these migrants [32]. Indeed, the translation and reproduction of "traditional foods" and their impact on health in migrant communities is subject to variation, depending on age at migration, household economic status, social context, and other variables [77].

The time, labor, and skill required for preparation of some traditional foods, such as *tejate*, may be a barrier to their preparation in Mexican immigrant households in the US, as mentioned. When adults have full time, low wage employment, time available for food preparation is limited, as is true for all full time wage earners in the US [78]. Although there is no systematic documentation, we estimated that, depending on the amount made, *tejate* preparation in Oaxaca takes at least two hours, not including the time to cook and clean the maize [33]. Time constraints and the trend some interviewees in Oaxaca described, of fewer young people learning the skill of *tejate* preparation, could lead to the loss of *tejate* and other traditional foods. However, it may also encourage the emergence of specialists, e.g., in SBQ *tejate* became available for sale in the public market in 2010 (Lucilia Martínez, SBQ farmer and *tejate* maker, personal communication September 2020), and recently there is a growing number of *tejateras* in greater LA. The skill, time, labor, and inaccessibility of ingredients may have contributed to the commercialization of powdered "instant *tejate*" such as "*Tejatli*" in the 2000s [32]. The accessibility, acceptability, and impact of instant *tejate* is unknown, beyond that it has not made *tejateras* obsolete.

### 5.2. Traditional Foods and Public Health

The complexity of discussing "traditional foods" with migrant communities may not be appreciated by health education programs, resulting in misunderstandings or inappropriate messaging. We suggest that qualitative transnational research can help avoid some of the obstacles to supporting migrant health [76], but will require collaborative, transdisciplinary research that uses community-based participatory research methods [79] to both accurately understand and reflect community experiences and values.

Mexican migrants in the US and their descendants have increased risk of diabetes and cardiovascular disease when compared to non-Hispanic whites [80]. There is a positive association between length of time in the US and negative health indicators, e.g., age-adjusted data for prevalence of obesity of Mexican migrants in the US was 28.1% for those in the US less for than 10 years, 40.1% for those in the US for more than 10 years, and 46.2% for Mexican Americans born in the US [81]. By maintaining a healthy and balanced diet, these NCDs can be prevented, and public health programs and interventions have expanded communication strategies to be accessible to Spanish-speaking communities and have been modestly effective [82]. However, there is still a lack of public health and dietary messaging promoting the consumption of the traditional Mexican foods and diets which have been associated with reduced risks of obesity, lower insulin levels, and reduced risks of prediabetes [83].

Currently, Los Angeles County has no public health policy or program that promotes migrants' traditional foods and diets, yet successful promotion of traditional foods has occurred at a large scale in the US. For example, the U.S. Department of Agriculture (USDA) launched the Indigenous Food Sovereignty Initiative in 2021 to promote "traditional food ways, Indian Country food and agriculture markets, and indigenous health through foods tailored to American Indian/Alaska Native dietary needs . . . partnering with tribal-serving organizations on seven projects to reimagine federal food and agriculture programs from an indigenous perspective" [84]. A previous public health intervention focusing on the cultural integration of traditional foods within this community to promote health and help prevent type 2 diabetes has reported that participants made healthier food choices post intervention [85]. With funding and policy support, similar public health programs could be executed in greater LA to educate and promote healthy habits and support the practice

of preparing and consuming traditional foods and beverages of migrant communities, such as *tejate*.

*5.3. Tejate, Maize Diversity, Farming, and Planetary Health*

*Tejate* ingredients other than maize, including those for *masa de pixtle* [33,35], have been imported into LA formally or informally from Oaxaca for at least the last 15 years. Immigrant households sometimes maintain ties to traditional flavors and foods through *paqueterías* or *envois*, a term for small packages of foods or ingredients from home regions and the services that transport them [76,86]. However, commercial, industrial maize, presumably from the US or Mexico, has been used [35], including maize grain sold commercially as bird feed [33]. But discerning Oaxacan consumers in California note the difference when compared to *tejate* made with Bolita maize, and efforts are underway to import the maize itself for making *tejate*, including by Zapotec *tejateras* in LA. An important area of future research will be to determine the amounts of traditional Oaxacan maize exported to California, or even grown by Oaxacan farmers in California, and the impact of these practices on maize diversity and farming, local diets, nutrition, and health across Oaxacalifornia.

As we have described elsewhere [33], the fate of the relationship between a traditional food and the crop diversity associated with it may follow different pathways (Figure 3), with implications for crop diversity and public and planetary health. Currently, for *tejate* and the maize used to make it in greater LA, pathway C is most common, but there are indications of some shift to pathway A due to the efforts of Oaxacans in both LA and Mexico. The form taken by such a change could affect its impact on planetary health by virtue of the location, scale, and form of maize production and its effect on the economic wellbeing of Oaxacan farming households and communities.

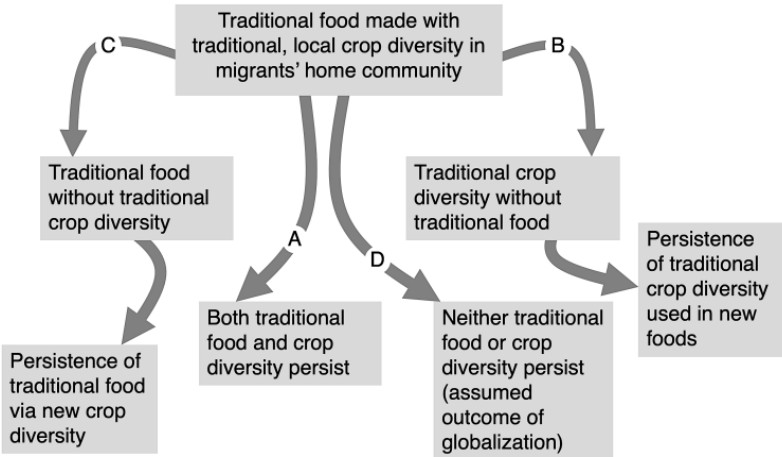

**Figure 3.** A conceptual framework for understanding the migration and transformation of traditional foods and crop diversity. A–D = pathways for the migration of traditional foods and related crop diversity. Based on [33], used with permission.

## 6. Conclusion: Public Health, Planetary Health, and *Tejate*

Based on current research, it is likely that attributes of *tejate* and other foam-topped Mesoamerican maize and cacao beverages, as well as other traditional foods, have the potential to make positive contributions to nutrition, health, wellbeing, crop diversity, and agroecology. As such they lie at the nexus of the many components of the public and planetary health crisis. As dynamic, valued elements of local communities, these foods may also provide a foundation for constructing effective, equitable responses to this crisis. Indeed, some scholars argue that inequity at many levels underlies the public and planetary health crisis (e.g., [87]). A collaborative, locally engaged investigation of the multiple impacts of traditional foods may provide an alternative to the market and technology-based approaches that are causing that crisis. Collaborative research is needed

for understanding, and to create consistent education and messaging that reflect both evidence and community values.

Residents of some Oaxacan communities have told us that their local health clinic advised against drinking *tejate*, declaring it unhealthy, especially because it could increase the risk for diabetes. Another study reported one SBQ resident as commenting that they had been discouraged from drinking *tejate* by the community health clinic [88]. The advice from health workers to avoid *tejate* is based on the well-established association between intake of SSBs and the risk of NCDs, discussed above. However, the extent of this public health guidance appears limited—no SBQ residents we interviewed described being discouraged from drinking *tejate*, and two people stated that they believed that they do not have diabetes because of their *tejate* consumption.

Based on current understanding of the potential positive impacts of *tejate* and other traditional maize beverages, there is a need for community–researcher collaboration such as community-engaged or community-based participatory research, to investigate: how to create holistic, culturally sensitive, and tailored health messaging around traditional foods; how to reduce negative health effects of traditional foods in the diet, and strengthen and increase their positive contributions to public and planetary health; and the effects of the changing cultural and social roles of traditional foods and beverages on crop diversity and the wellbeing of small-scale farmers. Public health policy and messaging that recognizes the diverse impacts of traditional foods may be an especially effective leverage point for supporting positive contributions to both public and planetary health in a transnational community such as Oaxacalifornia, and beyond.

**Author Contributions:** Conceptualization, D.S. and D.A.C.; investigation, D.S., D.A.C., F.A.C. and V.J.; writing—original draft, D.S., D.A.C. and F.A.C.; writing—review and editing, D.S., D.A.C., F.A.C., V.J. and M.C.W. All authors have read and agreed to the published version of the manuscript.

**Funding:** This research received no external funding.

**Institutional Review Board Statement:** This research was conducted in accordance with the Declaration of Helsinki and approved by the Institutional Review Board of the University of California, Santa Barbara (protocol code 4-01-250, record number 20090976, start date 1 September 2004).

**Informed Consent Statement:** This research was declared exempt by UCSB IRB.

**Data Availability Statement:** Data are available upon request from the first author.

**Acknowledgments:** We thank the households interviewed in Oaxaca; María del Carmen Castillo Cisneros for assistance with the Oaxaca interviews; Pulciano Gomez for orientation to SBQ tejateras in LA; *Grupo Folklórico Guish-Bac* for permission to use their image and discussions about *tejate* in LA; UCMexus-CONACyT for funding of the earlier research in Oaxaca discussed here.

**Conflicts of Interest:** The authors declare no conflict of interest.

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
