# Peer review of "Traditional Foods, Globalization, Migration, and Public and Planetary Health: The Case of Tejate, a Maize and Cacao Beverage in Oaxacalifornia"

_challenges, doi:10.3390/challe14010009_

Round 1

Reviewer 1 Report

Thank you for letting me review this interesting post.
1. the connections between the industrialization of agriculture and health problems are very well worked out. The connections to planetary health should be worked out in much more detail.
2. socio-cultural connections could be more summarized (shortened) here. On the other hand, the connections to the natural environment should be substantiated more strongly here as well. The statements in this section are mainly based on own, already published articles. It is not clear in the summary presented here what the methodology of the surveys was. Primarily quantitative data is presented in the table, especially for the interviews. I understand interviews as a method of qualitative research. Contrary to the quantitative data, the collection, processing and analysis of the qualitative data is not transparent and should be presented in the same way.
4.1 Here, geopolitical backgrounds of the large Mexican migration movements would be interesting and should be put into a context with changing dietary habits.
Overall, the article follows a comprehensible approach to the discussion of dietary habits and health. Unfortunately, the references to planetary health, as suggested in the title, are not addressed enough and should definitely be emphasized more.

Author Response

For the editor: We were not able to see the reviewers’ names, even though they requested to be identified

#1

Open Review

English language and style

( ) English very difficult to understand/incomprehensible
( ) Extensive editing of English language and style required
( ) Moderate English changes required
( ) English language and style are fine/minor spell check required
(x) I don't feel qualified to judge about the English language and style

Yes

Can be improved

Must be improved

Not applicable

Does the introduction provide sufficient background and include all relevant references?

( )

( )

(x)

( )

Are all the cited references relevant to the research?

( )

(x)

( )

( )

Is the research design appropriate?

( )

( )

( )

(x)

Are the methods adequately described?

( )

( )

( )

(x)

Are the results clearly presented?

( )

( )

(x)

( )

Are the conclusions supported by the results?

( )

( )

(x)

( )

Comments and Suggestions for Authors

Thank you for letting me review this interesting post.
1. the connections between the industrialization of agriculture and health problems are very well worked out. The connections to planetary health should be worked out in much more detail.
2. socio-cultural connections could be more summarized (shortened) here. On the other hand, the connections to the natural environment should be substantiated more strongly here as well. The statements in this section are mainly based on own, already published articles. It is not clear in the summary presented here what the methodology of the surveys was. Primarily quantitative data is presented in the table, especially for the interviews. I understand interviews as a method of qualitative research. Contrary to the quantitative data, the collection, processing and analysis of the qualitative data is not transparent and should be presented in the same way.

RESPONSE – We appreciate the request for more information regarding methods for qualitative research. We have clarified that the quantitative data presented is in fact from the interviews by adding “self reported” (e.g., L209ff) to explicitly communicate this. While interviews may be considered qualitative, they are often used to provide practitioner-reported quantitative data regarding practices and values that may be difficult to measure at a large scale, such as annual maize and tejate consumption for 75 households. Because this is a Perspective and not a research report, and for brevity, readers are referred to two peer-reviewed publications for further details, including regarding methodology.

4.1 Here, geopolitical backgrounds of the large Mexican migration movements would be interesting and should be put into a context with changing dietary habits.

RESPONSE – While we agree that the geopolitical factors related to or stimulating migration are part of the larger discussion of dietary change and diasporic populations, this is beyond the scope of the current Perspective piece. This Perspective highlights the specific, ethnographic, nutritional and agricultural implications of the persistence of a traditional food in one transnational community.

Overall, the article follows a comprehensible approach to the discussion of dietary habits and health. Unfortunately, the references to planetary health, as suggested in the title, are not addressed enough and should definitely be emphasized more.

RESPONSE – We have further elaborated on the planetary health implications of a shift to maize produced by industrial agriculture vs the traditionally-based, biodiverse systems most common in Oaxaca (e.g., L174-189, 415-422)

Submission Date

03 November 2022

Date of this review

11 Nov 2022 14:00:23 

Reviewer 2 Report

This article is focused on a relevant field – “dietary acculturation.

There are promising sections in the paper but there are also opportunities of improvement. Let me address the most claiming of these weak sections:

Point 1: The English style must be revised for the entire text. The Review of Literature and Motivation must be redone, considering a different path. For instances, I suggest Authors to divide the Review of Literature considering the different schools of the advantages of endogenous consumption, like the one based on tejate.

Authors must then elaborate and justify the rationale behind the set of variables/criteria suggested in the tested models.

Additionally, authors must discuss better their arguments. How can they claim so clearly “in addition to sugar, many of the calories in tejate are from starches in maize, a pro- 277
portion of which are resistant starch (RS)..” if not discussing the limitations of the observation techniques and of the possibility of alternative metrics? Lastly, There is not still a clear linkage between the Review of Literature and the empirical effort.

Authors must also consider the possibility of several other methods for robustness (mostly anticipated in the quoted literature – like questionnaires, interviews, longitudinal data, etc). Even the selection method of documents for the review of literature raised several issues, because this is missing the literature also discussing the weaknesses of endogenous diets (namely, the convenience of these diets to different life styles, the difficulty of following the original instructions and having the natural ingredients or even the risk of uncontrolled infection sources).

Point 2: Tables and Figures can be improved in the presented format. There are also different types/styles of edition which must be fixed.

There are additional insights which must be discussed again.

Point 3. Finally, I will also appreciate to have more information about the implications, weaknesses and challenges of the data sources and results.

Author Response

For the editor: We were not able to see the reviewers’ names, even though they requested to be identified

Open Review

English language and style

( ) English very difficult to understand/incomprehensible
( ) Extensive editing of English language and style required
(x) Moderate English changes required
( ) English language and style are fine/minor spell check required
( ) I don't feel qualified to judge about the English language and style

Yes

Can be improved

Must be improved

Not applicable

Does the introduction provide sufficient background and include all relevant references?

( )

( )

(x)

( )

Are all the cited references relevant to the research?

( )

(x)

( )

( )

Is the research design appropriate?

( )

(x)

( )

( )

Are the methods adequately described?

( )

( )

(x)

( )

Are the results clearly presented?

( )

(x)

( )

( )

Are the conclusions supported by the results?

( )

(x)

( )

( )

Comments and Suggestions for Authors

This article is focused on a relevant field – “dietary acculturation.”

RESPONSE – In this manuscript we start to construct a framework for understanding a transnational community and its traditional food undergoing modernization and globalization and how those relate to and interact with public and planetary health. We believe the brief discussion of dietary acculturation currently in the ms. is adequate. While dietary acculturation is relevant, the focus in our Perspective is other significant processes such as the resistance of diasporic communities to dietary acculturation by attempting to retain and recreate a version of traditional foods, and the significance of this for crop diversity, agricultural systems and wellbeing in CV Zapotec communities in Oaxacalifornia. Using the case of tejate, we provide an evidence-based reflection on how those phenomena and processes inter-relate, and how research may start to investigate that inter-relation at the local level, building on ethnography and collaboration, to assess how to strengthen public and planetary health. As in our response to the first reviewer, we believe it is helpful to reiterate that this is a “Perspective” manuscript and not a Research or Review article. We discuss literature to place the discussion in context, and provide evidence and further resources for statements in the manuscript, however this is not intended to be an exhaustive review of the literature.

There are promising sections in the paper but there are also opportunities of improvement. Let me address the most claiming of these weak sections:

Point 1: The English style must be revised for the entire text. The Review of Literature and Motivation must be redone, considering a different path. For instances, I suggest Authors to divide the Review of Literature considering the different schools of the advantages of endogenous consumption, like the one based on tejate.

RESPONSE – We have reviewed the manuscript and made the language as clear as possible; 4 of the 5 authors are either native English speakers and writers, or have been for decades, and we do not find any significant stylistic issues. The point of considering negative as well as positive characteristics of traditional foods is well taken; and has been our intent, as indicated in the last line of the of the abstract. Our discussion of the use of sugar in tejate is recognition of the negative individual and public health consequences of some traditional foods that must be addressed if their benefits are to be realized (e.g., L284-288, 298ff). We have also added  text to reflect the role of time and labor intensive preparations that may affect consumption of such foods, but also may explain the growing presence of specialist producers servicing their own migrant community, see L354ff.

Authors must then elaborate and justify the rationale behind the set of variables/criteria suggested in the tested models.

RESPONSE –  We disagree with this comment. In both the literature cited, and beyond, the variables considered in our Perspective are recognized as central to food sovereignty and cultural identity (traditional recipes with authentic ingredients and access to these), individual and public health (major non-communicable diseases, sociocultural and economic wellbeing), planetary health (low input, diverse, small scale agriculture). Additionally, because our ms is a Perspective, not a research or review ms., we do not believe it necessary to further discuss the rationale for the variables and criteria in the models.

Additionally, authors must discuss better their arguments. How can they claim so clearly “in addition to sugar, many of the calories in tejate are from starches in maize, a pro- 277
portion of which are resistant starch (RS)..” if not discussing the limitations of the observation techniques and of the possibility of alternative metrics? Lastly, There is not still a clear linkage between the Review of Literature and the empirical effort.

RESPONSE – We agree with the reviewer that this quote is inappropriate as an unsubstantiated assertion. We have corrected this because the quote is not our claim, but rather evidence from laboratory studies whose data we included in Table 2, along with appropriate citations. We have referred readers to the table after the statement listed by the reviewer. While we appreciate the point that alternative metrics could and in the future should be applied to assessing tejate’s relationship to individual and public health, the significance of type 2 diabetes in the global public health crisis, and especially in Mexico, with the highest rate of diabetes incidence and mortality globally, makes understanding the beverage’s sugar content in particular very important. That evidence is cited and the table constructed in part to highlight the contrast between SSB energy content – entirely from sugars – and that of tejate much of which is from starches including resistant starches which appear to reduce glycemic response.

Authors must also consider the possibility of several other methods for robustness (mostly anticipated in the quoted literature – like questionnaires, interviews, longitudinal data, etc). (A) Even the selection method of documents for the review of literature raised several issues, because this is missing the literature also discussing (B) the weaknesses of endogenous diets (namely, the convenience of these diets to different life styles, the difficulty of following the original instructions and having the natural ingredients or even the risk of uncontrolled infection sources).

RESPONSE A – Because this is a Perspective and not a Review article, we did not do a systematic review of the literature for the wide range of topics covered, or a comparative analysis of the quality of research methods used beyond ensuring the work was published in reputable peer reviewed journals and used appropriate, documented methods. However, we did do an informal, comprehensive search for all sources on tejate, and have cited that extensively. It includes our own research using interviews, questionnaires, archeological and biochemical methods. However, we have also cited other works such as:

(Esparza 2017, 2018, González Esperón 2006, González-Amaro et al. 2015, Martinez 2008, Rivera 2019)

Esparza B. 2017. Taste History With Tejate, an Ancient, Pre-Hispanic Street Drink. EATER LA. Los Angeles, CA.

---. 2018. Tejate: The Most Magical Drink in Oaxaca. EATER LA. Los Angeles.

González Esperón LM. 2006. El Tejate: Una Bebida Prehispánica. Secretaría de Cultura del Estado de Oaxaca.

González-Amaro RM, de Dios Figueroa-Cárdenas J, Perales H, Santiago-Ramos D. 2015. Maize races on functional and nutritional quality of tejate: A maize-cacao beverage. LWT-Food Science and Technology 63:1008-1015.

Martinez R. 2008. Inició con 5 kilos de téjate; hoy produce una tonelada. Noticias-Voz e Imagen de Oaxaca. 2008 March 12.

Rivera S. 2019. El téjate, una bebida oaxaqueña que le apuesta no solo a ‘milenios’ sino también a los anglosajones. Los Angeles Times. 2019 August 29.

RESPONSE B– Thank you for the excellent point regarding balanced consideration of traditional foods. We have stated the skill and labor demands of tejate preparation more explicitly, including their interpretation in archeological contexts (L146-150; 160-ff). We revised the discussion to provide a more balanced overview of factors necessary to consider in collaborative evaluation of the positive and negative aspects of traditional foods, including time, labor and skills required for their preparation, and how this appears to be motivating specialists to provide commercial production in both Oaxaca and LA (see L354ff).

Point 2: Tables and Figures can be improved in the presented format. There are also different types/styles of edition which must be fixed.

There are additional insights which must be discussed again.

RESPONSE – We have reviewed the tables and figures; we’ve replaced Fig 3 with a higher quality image, and did not find problems with the other two figures. All of the tables have been formatted according to Challenges’ templates and we find no additional problems.

Point 3. Finally, I will also appreciate to have more information about the implications, weaknesses and challenges of the data sources and results.

RESPONSE –Again, we feel further discussion of methods is neither necessary or appropriate as we are citing peer reviewed work, and this is a Perspective piece in which we explore the construction of a framework for bringing traditional foods, public and planetary health together, using the example of tejate

Submission Date

03 November 2022

Date of this review

28 Nov 2022 09:42:47

Round 2

Reviewer 1 Report

Thank you for this study on a very specific and controversial topic. Overall, the impact on the environment could be considered in more depth empirically in the following research projects.

Reviewer 2 Report

As this is a piece of Perspective, I think it can be published in Challenges (however, I must state the Responses were weak and they should have influenced my decision against the publication if this was a Research paper).